# Protocol for Rhapsody: a longitudinal observational study examining the feasibility of speech phenotyping for remote assessment of neurodegenerative and psychiatric disorders

Elliot Hampsey  ,[1] Marton Meszaros,[2] Caroline Skirrow,[2] Rebecca Strawbridge,[1] Rosie H Taylor,[1] Lazarus Chok,[2] Dag Aarsland,[1] Ammar Al-Chalabi  ,[1] Ray Chaudhuri,[1,3] Jack Weston,[2] Emil Fristed,[2] Aleksandra Podlewska,[1,3] Olabisi Awogbemila,[3] Allan H Young[1]

EH and MM contributed equally.

[1]Institute of Psychiatry, Psychology, & Neuroscience, King's College London, London, UK
[2]Novoic Limited, London, UK
[3]Parkinson's Foundation Centre of Excellence, King's College Hospital NHS Foundation Trust, London, UK

**Correspondence to**
Elliot Hampsey;
elliot.hampsey@kcl.ac.uk

## STRENGTHS AND LIMITATIONS OF THIS STUDY

⇒ Remote, partially self-administered speech-battery enabling use by those with difficulty accessing in-person healthcare.
⇒ Speech battery and study design uniquely allows for investigation of the same tasks across a wide number of indications.
⇒ Repeated assessment allows for examination of test reliability, and intrasubject and intersubject variability across key clinical groups.
⇒ Limited sample size.
⇒ Short follow-up period limits scope for longitudinal disease monitoring.

## ABSTRACT

**Introduction** Neurodegenerative and psychiatric disorders (NPDs) confer a huge health burden, which is set to increase as populations age. New, remotely delivered diagnostic assessments that can detect early stage NPDs by profiling speech could enable earlier intervention and fewer missed diagnoses. The feasibility of collecting speech data remotely in those with NPDs should be established.

**Methods and analysis** The present study will assess the feasibility of obtaining speech data, collected remotely using a smartphone app, from individuals across three NPD cohorts: neurodegenerative cognitive diseases (n=50), other neurodegenerative diseases (n=50) and affective disorders (n=50), in addition to matched controls (n=75). Participants will complete audio-recorded speech tasks and both general and cohort-specific symptom scales. The battery of speech tasks will serve several purposes, such as measuring various elements of executive control (eg, attention and short-term memory), as well as measures of voice quality. Participants will then remotely self-administer speech tasks and follow-up symptom scales over a 4-week period. The primary objective is to assess the feasibility of remote collection of continuous narrative speech across a wide range of NPDs using self-administered speech tasks. Additionally, the study evaluates if acoustic and linguistic patterns can predict diagnostic group, as measured by the sensitivity, specificity, Cohen's kappa and area under the receiver operating characteristic curve of the binary classifiers distinguishing each diagnostic group from each other. Acoustic features analysed include mel-frequency cepstrum coefficients, formant frequencies, intensity and loudness, whereas text-based features such as number of words, noun and pronoun rate and idea density will also be used.

**Ethics and dissemination** The study received ethical approval from the Health Research Authority and Health and Care Research Wales (REC reference: 21/PR/0070). Results will be disseminated through open access publication in academic journals, relevant conferences and other publicly accessible channels. Results will be made available to participants on request.

**Trial registration number** NCT04939818.

## INTRODUCTION

### Healthcare costs of neurological and psychiatric disorders

Neurological and psychiatric disorders (NPDs) affect 20% of older people,[1] at an estimated cost to the UK of >£68 billion annually.[2–5] With over 65s estimated to make up over a quarter of the UK population by 2050,[6] the burden of NPDs on society will increase dramatically. Alzheimer's disease accounts for 50%–70% of dementia cases, with an estimated £34.7 billion annual healthcare cost in the UK.[2] Dementia with Lewy bodies (DLB) accounts for 15%–20% of all dementia cases,[7–9] affecting >100 000 people in the UK. DLB is a much more rapidly progressing disease, with a median survival of 3.72 years postdiagnosis.[10]

While less prevalent than dementias, Parkinson's disease (PD) and motor neuron disease (MND) confer extensive costs due to high morbidity and mortality. PD affects >160 000 people in the UK,[11] costing the National Health Service (NHS) >£2.2 billion annually,[12] and over 80% may progress to dementia in the long term while the rate is higher in those with the cholinergic subtype.[13] MND affects about 5000 people in the UK,[14] with patients suffering severe impairments in addition to considerable lifespan reduction.[15]

Although affective disorders typically onset during one's late 20s,[16] they are also marked by recurrence throughout the lifespan.[17] WHO reports major depressive disorder (MDD) to have the highest disability burden of all conditions internationally,[18] incurring a care cost in the UK of >£23.8 billion annually.[4] Although less prevalent (1%–4.5% lifetime prevalence),[19 20] the burden of bipolar disorder remains considerable, costing the NHS £1.6 billion annually.[5] People with bipolar disorder have an average of 10–20 years reduced lifespan compared with the general population, due in part to the 20% of people with bipolar disorder who die by suicide.[21]

## Diagnostic challenges

NPDs are complicated by challenges to accurate and early diagnosis. Estimates suggest that approximately 50% of depressive episodes go undetected,[22] while >60% of MDD and >90% of patients with bipolar disorder may be misdiagnosed.[23] Similarly, ≥50% of those who show evidence of DLB post mortem were not diagnosed with the disease during life,[24] with one study finding that 39.5% of those diagnosed clinically as free from Alzheimer's disease (AD) met the minimum histopathological threshold.[25] High rates of diagnostic errors in bipolar disorder lead to the delaying of a correct diagnosis by 5.7 years,[26] with many waiting for more than a decade.[27] Ostensibly, mis/delayed diagnosis can impair treatment provision, exacerbate course of illness, limit treatment options and reduce quality of life.[28]

Overlap between the clinical presentation of NPDs makes diagnosis challenging. For example, cognitive impairment, the hallmark symptom of prodromal AD, is also seen in all patients with DLB[29] as well as in >80% of patients with PD across all motor stages of the condition[30] and is common in people with depression. The cognitive assessments used to screen for dementia are inadequate, leaving 32% of patients with early stage AD[31] and 50% of patients with DLB[32] undiagnosed. Diagnosing other neurodegenerative disorders such as PD is particularly challenging, as there are no definitive tests.[33] Physicians instead rely on often error-prone clinical judgement,[34] with an estimated 20% of those with PD who have come to medical attention going undiagnosed.[35]

Even when diagnoses are made correctly, this can be years after symptoms have begun. Failure to intervene early is associated with a more severe impact on quality of life, such as memory, motor and psychiatric disturbances.[36–39]

## Digital and remote assessment strategies

Uptake of digital and remote assessments methods has accelerated during the SARS-CoV-2 global pandemic, both in research and clinical practice.[40 41] Digital health technologies hold promise for reducing burden and improving access for those who travel to medical centres would be laborious, stressful or financially challenging.[42]

Digital technology can help to enhance certain aspects of assessment practices. Higher frequency assessment, with automated administration and completed remotely, can allow for more detailed assessment of behaviour and cognition over time. Higher frequency remote assessments have been successfully deployed in mood disorders and other psychiatric conditions,[43 44] and in neurological conditions, including mild cognitive impairment (MCI), mild AD[45] and PD.[46 47] These studies also typically report moderate to high levels of adherence to remote assessment and good acceptability of this method of assessment. Furthermore, digital speech capture can help to enrich analyses with more advanced text similarity analyses[48] and automated extraction of language features commonly evaluated during connected speech,[49] and furthermore incorporate vocal and acoustic features, which are sensitive to clinical status.[50–53]

While holding promise for improving convenience and access, there is concern regarding whether digital assessment is particularly challenging for certain populations, for example, in those with dementia or cognitive impairment.[54] Data integrity may be compromised by participants misunderstanding instructions (which are not or cannot be prompted for correction), or due to environmental/contextual limitations, such as distractions or ambient noise.[47] Additional care is required in the analysis of high-frequency assessments, which needs to take into account the autocorrelation of repeated observations within individuals, and potential variation within individuals over time.[47]

## Novel strategies to improve illness recognition

Artificial intelligence-based techniques have shown efficacy in analysing medical data in diagnosis and could potentially detect and translate subtle, early changes in speech into predictive diagnostic models.[55] This would assist care providers and their patients, who are likely to benefit from objective profiling biomarkers that are capable of disambiguating clinically similar NPDs. Speech/Language alterations in NPDs are promising universal biomarkers as they reflect subtle cognitive, motor and mood changes.[56–59] As speech is among the first modalities to be affected in NPDs,[56 57 60–62] developing speech/language-based profiling biomarkers will also aid in early diagnosis of NPDs. Herewith, we describe the research protocol of the 'Rhapsody' study, funded by the National Institute for Health Research, which aims to assess the feasibility of recording and detecting changes in patterns of speech across a range of high burden NPDs.

## Study objectives

The primary objective of the study is to evaluate the feasibility of eliciting continuous narrative speech to collect speech data remotely in three groups of NPDs: neurodegenerative cognitive disorders, other neurodegenerative disorders (PD and MND) and affective disorders. We hypothesise that due to the simple speech-based interface, participants will be able to engage with and provide speech data during virtual study visits and remote assessments. Basic feasibility will be assessed via the average length of speech elicitation for each speech task (in seconds) during the first week of self-assessment. Additionally, we will examine the proportion of participants in whom narrative speech lasting at least 20s is elicited within each group.

The study also has secondary objectives concerning the feasibility of using speech tasks to collect speech data in the remote setting, such as evaluating reliability of repeated assessments across related (comparing virtual visits vs fully remote assessments) or repeated tasks (parallel variants administered across days), examining intra-individual and inter-individual variance. We will also evaluate adherence to daily remote assessments, and participant-rated usability of the remotely administered application, measured via a brief usability questionnaire completed on the app.

Furthermore, the study will examine whether acoustic and linguistic patterns can be used to distinguish from group-specific control participants, and from other clinical indications, and whether this is impacted by relevant contextual or disease information covariates.

## METHODS AND ANALYSIS

### Design

The RHAPSODY study is an observational longitudinal study during which participants remotely complete a set of tasks designed to elicit speech at baseline and during the 4-week follow-up period. Participant-rated questionnaires of clinical symptoms are completed at baseline and repeated at follow-up weeks 2 and 4.

### Participants

Potential participants will be identified through local clinical services, cohort-specific internal databases, advertisements on a variety of social media platforms and dedicated websites for research participation. Participant recruitment began on 15 July 2021 and will continue to 30 June 2022.

Participants will meet group-specific eligibility criteria for neurodegenerative cognitive disorders (n=50), or other neurodegenerative motor disorders (n=50), or affective disorders (n=50), in addition to matched controls (n=75). The sample size was based on a review of feasibility study group sizes.[63] Individuals must be native English speakers, have access to a smartphone and personal computer/laptop which can connect to the internet and is capable of audio recording as well as

running an operating system of macOS X with macOS 10.9 or later; or Windows 7 or above. Exclusion criteria include: current substance use disorder, stroke within the last 2 years, transient ischaemic attack or unexplained loss of consciousness within the last 12 months or current risk of suicide. The additional group-specific eligibility criteria are as follows:

Group 1: neurodegenerative cognitive disorders (n=50): participants must be aged 50–85 years, with a score of 20–30 on The Telephone Interview for Cognitive Status-modified (TICS-M)[64] and one of the following clinical diagnoses (within the last 5 years):

A. MCI due to AD or mild Alzheimer's dementia as per the National Institute of Aging—Alzheimer's Association core clinical criteria (2011).[65]

B. Probable or possible DLB as per The Dementia with Lewy Bodies Consortium definition.[66]

C. Cognitive impairment (not due to AD/DLB), either diagnosed with behavioural variant frontotemporal dementia,[67] semantic variant primary progressive aphasia or non-fluent variant primary progressive aphasia,[68] vascular dementia[69] or unspecified MCI.

Group 2: other neurodegenerative disorders (n=50):

A. Participants with MND must be aged 18–85 years and score stage 3 or less on the King's amyotrophic lateral sclerosis staging system.[70]

B. Participants with PD must be within 5 years of a diagnosis of idiopathic PD according to the UK Brain Bank Criteria, aged between 30 and 85 years, and have a Hoehn and Yahr stage[71] of ≤2.

Group 3: affective disorders (n=50): participants with affective disorders must be aged 18–85 years, diagnosed with either major depression or bipolar disorder, in a current depressive episode according to the Diagnostic and Statistical Manual of Mental Disorders, fourth edition (DSM-IV) criteria as assessed by the Mini-International Neuropsychiatric Interview (M.I.N.I.) V.5.0,[72] of at least moderate severity as assessed on the Clinical Global Impressions Scale.[73]

Group 4: unaffected controls (n=75): approximately 25 unaffected control participants will be recruited for each cohort, matched for gender, age and education levels. Participants will be in otherwise good health; they may experience mild disorders (of metabolic, respiratory, immunological, cardiologic and metabolic origin) that do not impair daily functioning.

### Data collection procedures

Study procedures are summarised in figure 1. Potential participants will be provided with a patient information sheet by the research team, to be followed by the study visit ≥24 hours later. Participants providing consent will complete the study visit via video conferencing software. Participants will first complete the screening and baseline assessment phases, which are completed in a single sitting.

During the screening phase, relevant demographic information, medical history and information on current

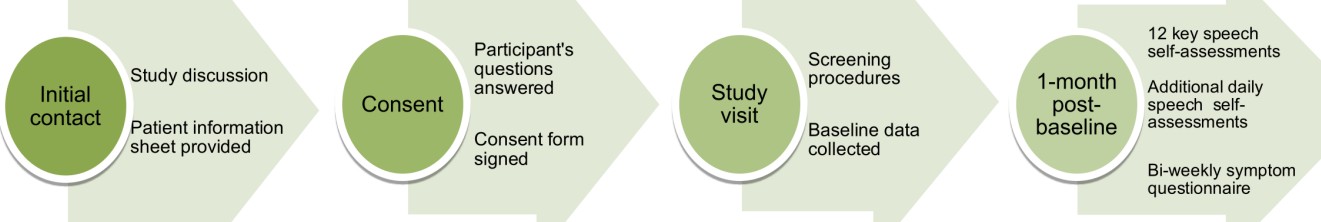

**Figure 1** Flow diagram of the participant journey in the RHAPSODY study.

medications are obtained, as well as ruling out visual or hearing difficulties that would serve as exclusion criteria. Some cohort-specific assessments (table 1) will also be used to determine eligibility. Participant eligibility is then evaluated against inclusion/exclusion criteria, with ineligible individuals exiting the study at this stage. Those who are eligible to continue will then complete the baseline data collection phase during the same teleconference.

A. (All groups): the Patient Health Questionnaire-9 item[74] is a short, self-report tool which uses a 4-point Likert scale to assess nine items relating to depressive symptomology.

B. (Group 3): M.I.N.I. is a structured interview for mental health diagnoses according to the DSM-IV criteria. The suicidality scale will be completed by any participant scoring ≥10 on the Patient Health Questionnaire-9 item.

C. (Group 3): the Clinical Global Impressions Scale is a 3-item, researcher-administered tool that provides an overall 'global' judgement concerning the participant's severity of depressive illness.

D. (Group 1): TICS-M is a short, researcher-administered assessment of cognitive functioning validated for use over the phone.

E. (Group 3): the Inventory of Depressive Symptomatology (30-item Clinician Rated)[75] is a researcher-rated instrument that provides detailed assessment of depression severity over the previous 7 days.

### Verbal cognitive and speech tasks

All participants will complete the same sequence of verbal cognitive and speech assessments. An overview of speech tasks is provided in table 2. Some of the tasks are optimised to elicit continuous narrative speech, such as reading text and recalling a presented story. Others elicit specific categories of words, or repetition of sentences or sounds.

These tasks combined allow for evaluation of a range of cognitive, linguistic and acoustic features elicited during spoken responses. Differential impairments are expected across groups and tasks, for example, while neurodegenerative cognitive disorders such as AD are

| Study procedure | Screening/Baseline | Week 1 | Week 2 | Week 3 | Week 4 |
|---|---|---|---|---|---|
| Medical history, concomitant medication overview and demographics | X | | | | |
| Telephone Interview for Cognitive Status-modified (group 1 only) | X | | | | |
| The Mini-International Neuropsychiatric Interview* (psychiatric diagnostic evaluation) (group 3 only) | X | | | | |
| Clinical Global Impression (group 3 only) | X | | | | |
| Speech/Language assessments (see table 2) | X | X | X | X | X |
| Inventory of Depressive Symptomatology—Clinician Rated (group 3 only) | X | | | | |
| Patient Health Questionnaire-9 item† | X | | X | | X |
| Generalised Anxiety Disorder Assessment-7 item† | X | | X | | X |
| Maudsley 3-Item Depression Visual Analogue Scale† | X | | X | | X |
| Altman Self-Rating Mania Scale† (groups 2 and 3 only) | X | | X | | X |

**Table 1** Screening tools and questionnaires completed at baseline and follow-up

(Self-assessments span Week 1–Week 4)

*Suicide questionnaire administered if participant scores >10 on the Patient Health Questionnaire-9 item.
†Patient-rated.

**Table 2** Overview of speech tasks used in RHAPSODY (main and/or remote assessments)

| Speech and cognitive tasks | Task details | Task characteristics and selected evidence from prior research |
|---|---|---|
| Count 1–10 or 10–1 | Participants are recorded counting from 1 to 10 or from 10 to 1. | The counting task was used as a brief familiarisation task for participants. |
| The Automatic Story Recall Task (ASRT) with immediate and delayed recall | The ASRT has 36 parallel story variants (18 'short' and 18 'long' stories), presented at a steady reading rate and matched for linguistic and discourse measures, including number of words, number of sentences, number of dependent clauses, mean sentence length and ratio of dependent clauses to t-units.[45] Prerecorded stories are played to the participants, who are be asked to recall the stories in as much detail as possible immediately and again after a delay. | Story recall is often used to evaluate verbal episodic memory. Prior meta-analytic evidence suggests that lesser impairment would be expected in story recall performance for individuals with DLB than those with AD-MCI.[91] A smaller head-to-head research study indicates that lesser impairment may be expected for individuals with PD than AD-MCI.[92] Meta-analytic evidence suggests that the task may be sensitive to mood disorders, with individuals with a history of depression having task performance deficits in story recall, with large effect sizes reported.[93] |
| Category fluency task | Participants will be given a category and required to give as many examples of that category as they can in 60s. Participants will complete parallel variants of this task, commonly implemented in the research literature, including animals, vegetables, fruits. | Category fluency evaluates semantic memory function and taps into attentional and executive functions. Impairments in category fluency have been noted in a rage of the clinical indications, including MCI/AD and DLB,[94] PD,[93] MND,[95] MDD[93] and BD,[96] but with different reported effect sizes. |
| Letter fluency task | Participants will be given a letter of the alphabet and required to give as many examples of words beginning with that letter as they can in 60s. Letters prompted include F, A and S. | Letter fluency tasks evaluate lexical access through the phonological route and are believed to more strongly depend on executive functions than category fluency measures. Prior research shows that task performance is impaired in adults with MCI,[97] PD,[94] MND,[95] following MDD[93] and in BD,[96] but with different reported effect sizes. |
| Action fluency task | Participants are tasked with naming as many of examples of things that 'people do', that is, generate verbs such as 'run' or 'work', in 60s. | Impairments in verb fluency tasks are thought to more greatly reflect frontostriatal neuropathology and neurochemical deterioration known to occur with progression of PD.[98] A systematic review[99] reports more prominent impairment for action fluency than for category fluency for PD, differential task performance for AD and DLB participants in action fluency and no difference in action fluency for comparisons of MCI and healthy control participants. |
| Digit span forwards and backwards | In this abbreviated digit span task, in forwards digit span, a series of 5 digits is presented to participants (eg, 8-3-1-9-6), which they are asked to repeat back in the same order. In backwards digit span, a series of 5 digits is presented to participants (eg, 8-3-1-9-6), which they are asked to repeat in backwards order (eg, 6-9-1-3-8). Forwards and backwards digit span were completed three times each with three different 5-digit sequences. | Digit span tests are associated with auditory attention and short-term memory function, with greater reliance on working memory for the backwards span variant. Meta-analyses show greater impairments in backwards span than in forwards span for MCI/AD[91]; PD.[100] In affective disorders, meta-analyses have emphasised deficits in backwards span in the absence of deficits in forwards digit span.[77 93] |
| Stroop | In this abbreviated Stroop task, participants are first presented with a grid of 50 colours written out in text (eg, 'BLUE') and asked to read these back as quickly as they can. They are then presented with a panel of colours presented in blocks and are asked to name these as quickly as possible. Finally, they are presented with a panel of 50 words typed with the typeface in a colour incongruent with the written word (eg, 'GREEN' written in the colour red). They are asked to state the colour of the letters as quickly as possible. | The Stroop test is an extensively used test to evaluate inhibition to verbal interference. Prior meta-analyses have shown impairments in Stroop interference performance in individuals with amnestic MCI[101]; PD[94]; MND[95]; MDD[77] and BD.[78] |
| Procedural discourse questions | Participants are tasked with describing, in as much detail as possible, how they would go about doing the dishes by hand (in main assessment), how they would go about posting a letter and how they would make a cup of tea (each assessed on one occasion during remote assessments). | The procedural discourse task elicits naturalistic speech, used to express temporal and hierarchical steps in a behavioural sequence. The task has been found to be sensitive to speech differences in individuals with MCI and mild AD in comparison with healthy control participants.[102] |
| Picture description task | Participants verbally describe a picture in as much detail as they can. Four different pictures will be used, the first being the 'cookie theft' picture taken from the Boston Diagnostic Aphasia Examination battery[103] administered during main assessment, and the other two being the 'rescue' picture[104] and two simple emotion eliciting pictures administered during remote assessments.[105] | The picture description task produces a structured output that can be scored according to the completeness of response, and is useful for measuring and detecting differences in language content, syntax, pragmatics and acoustic features.[49] Research shows changes in this task for individuals with MCI and AD dementia, lesser impairment in individuals with depression, and differences in speech error corrections between AD and PD groups.[49] |

**Table 2** Continued

| Speech and cognitive tasks | Task details | Task characteristics and selected evidence from prior research |
|---|---|---|
| Sequence narration task | Participants must describe their narrative interpretation of a series of images presented as a storyboard, taken from the 'Argument' sequence.[104] Participants have 1 min to look at the picture sequence and are then asked to tell the story represented by the picture sequence. | The task produces a relatively structured output, where analysis outputs can include completeness of the narrative. The task is also used to evaluate lexico-semantic deficits, and syntactic complexity. The pictorially provided structure is thought to place a decreased load on working memory.[106] |
| Reading a script | Participants will read a short passage aloud: 'The boy who cried wolf'.[107] A passage designed for phonetic description and acoustic research on varieties of English. | This task allows the evaluation of pronunciation variations in the English language in different accents and dialects. |
| Sustained phonation task | Participants are required to phonate several sounds (/a:/, /i:/ and /u:/) for as long and steadily as possible, in one breath. | Sustained phonation allows for the measurement of voice quality. Changes in vowel articulation has been noted as a potential early marker for PD, and research shows changes in the audio speech measures distinguish PD participants from controls and a likely increase in voice impairment following developmental trajectory of the individuals. |
| Sentence reading task | During sentence repetition, a sentence (containing 11 or 15 words) is presented auditorily to the participant is asked to must repeat it back in the same way and in one breath. The sentences each contain two /u:/, two /i:/ and two /a:/ long corner vowels to measure vowel space area, following voiceless consonants. During each assessment, two different sentences are presented. | Sentence repetition allows for the measurement of voice quality within a more naturalistic speech response. Research indicates that sentence repetition may be more sensitive to audio speech changes in PD than sustained phonation.[108] |
| Syllable repetition task | Participants are required to repeatedly phonate a syllable as quickly as possible for one breath. They are first asked to compete this for the syllable /ba/ and then for three consecutive syllables /pa/ /ta/ /ka/. | These tasks assess difficulties with phonology, articulation and working memory. Prior research shows changes in acoustic features in the /pa/ /ta/ /ka/ task in PD (including jitter, pause rates, intensity variation, alternating motion rate and pitch variation).[108] Other research shows changes acoustic features (including alternating motion rate, jitter, frequency and an overall variable) for MND in repeated phonation of the syllable /ba/.[109] |

AD, Alzheimer's disease; BD, bipolar disorder; DLB, dementia with Lewy bodies; MCI, mild cognitive impairment; MDD, major depressive disorder; MND, motor neuron disease; PD, Parkinson's disease.

commonly associated with anomia and decreased information content, other neurodegenerative conditions such as PD show greater impairment in volume and articulation.[53 56] Language-based changes have also been noted in PD including rates of pauses, phrase length and changes in sentence generation and construction.[76] Similarly, cognitive and speech changes in mood disorders are commonly described.[52 77–79]

The consistent use of tasks across groups allows for a head-to-head comparison of task performance and speech measures for different clinical indications and different tasks. The tasks are selected due to a rich research background showing their involvement in specific clinical indications (described in table 2). However, with a few notable exceptions, cross-diagnostic evaluations are not commonly reported.

Speech tasks will be administered and recorded using the application, with all participants completing the same set of speech tasks. Participants will be supported with using the Novoic application on their smartphone. Audio recordings will be transferred to a secure server and then deleted from the smartphone device.

Recordings of the combined screening and baseline visit will also be made simultaneously by video conferencing software. Zoom (https://zoom.us) will be used due to its ability to turn off audio manipulation effects. The main assessment phase begins with a series of speech tasks, self-recorded using a smartphone application,

that the researcher guides the participant to complete. An overview of the specific tasks completed in the main assessment is provided in table 3.

Following the completion of the main sequence speech tasks, all participants will then be asked to complete online self-rated questionnaires via the Qualtrics platform (www.qualtrics.com) (see table 1):

A. The Generalised Anxiety Disorder Assessment-7 item[80] is a brief tool for assessing symptoms of anxiety. Participants rate seven items relating to core symptoms of anxiety on a 4-point Likert scale concerning the frequency at which those symptoms occur.

B. The Maudsley 3-Item Depression Visual Analogue Scale[81] is a researcher-rated assessment of depressive illness which detects symptom severity and suicidality.

C. The Altman Self Rating Mania Scale[82] is a 5-item instrument designed to self-assess the presence and severity of manic and hypomanic symptoms by comparing how they feel currently to their non-affected baseline state.

Participants will then complete follow-up data collection over a period of 4 weeks using the smartphone application. Participants will be notified via the mobile application to complete 12 brief (approximately 15 min) unsupervised speech and cognition assessments over the month following the baseline visit. Participants will receive a different set of speech tasks each day from the application's built-in task-bank, including the same main

**Table 3** Assessments completed during main (supervised) assessment, and remotely via the application on the participants' own smartphones

| Task | Main (supervised) assessment | Remote assessment day | | | | | | | | | | | |
|---|---|---|---|---|---|---|---|---|---|---|---|---|---|
| | | 1 | 2 | 3 | 4 | 7 | 8 | 9 | 10 | 11 | 14 | 21 | 28 |
| Counting 1–10 or 10–1 | X | X | X | X | X | X | X | X | X | X | X | X | X |
| Automated Story Recall Task immediate and delayed recall | X | X | X | X | X | X | X | X | X | X | X* | X* | X* |
| Category fluency | X | X | | | | | X | | | | X* | X* | X* |
| Letter fluency | X | | X | | | | | X | | | X* | X* | X* |
| Action fluency | X | | | | | | | | | | X* | | |
| Digit span forward and backward | | | | X | | | | | X | | | | |
| Stroop | | | | | | | X | | | | | | |
| Procedural discourse | X | | | | X | | | | | X | | | |
| Picture description | X | | X | | | | | X | | | | | |
| Sustained phonation | X | | | | | | | | | | X* | | |
| Syllable repetition | X | X* | | | | | | | | | X* | | |
| Sentence repetition | X | X* | | | | | | | | | X* | | |
| Sequence narration | X | | | | | | | | | | | | |
| Reading a script | X | | | | | | | | | | | | X* |
| Usability questionnaire | X | | X | | | | | | | X | | | |
| Vision and hearing questionnaire | X | | | X | | | | | | | | | |
| Baseline mood, sleep, attention and effort questionnaire | X | X | | | | | | | | | | | |
| Daily mood, sleep, attention and effort questionnaire | X | X | X | X | X | X | X | X | X | X | X | X | X |

Remote assessment day corresponds to the number of days after the virtual visit. Test order is shown in descending order. All Automated Story Recall Task immediate and delayed recall assessments are completed with brief distractor tasks occurring in-between.
*Repeated stimuli.

speech and cognition tasks described above, and abbreviated test of executive function (Stroop, digit span).

Additional, brief self-report measures, incorporated into remote assessments, included:

► A 4-item questionnaire on the usability of the smartphone application.
► A brief, 3 min long questionnaire on whether the participant has any visual or hearing impairments, and whether they had related difficulties with the administered tasks.
► A baseline 4-item questionnaire: examining typical mood, sleep, mind wandering and effort.
► A brief, daily 4-item questionnaire on current state: mood, sleep, mind wandering and effort.

The remote assessment schedule is provided in table 3. Participants will also complete follow-up questionnaires online via Qualtrics during weeks 2 and 4, which are shown in table 1.

## Statistical analysis plan

The main analysis objectives of the study include examining: (1) feasibility of eliciting and collecting speech data within different clinical groups, (2) reliability, and intrasubject and intersubject variance of speech task performance, (3) adherence to remote unsupervised assessments, (4) app usability and (5) to evaluate whether acoustic and linguistic patterns can be used to distinguish between clinical indications and controls.

Feasibility will be assessed as the length of speech generated during speech elicitation tasks. The primary end point is calculated as the number of participants who successfully complete Automated Story Recall Task (ASRT) assessments as a fraction of the disease cohort, which may indicate likely suitability of this procedure within each disease group. Participants are considered successful where at least one of the immediate story recalls produces a spoken response spanning ≥20 s from the first to the last word. The effect of demographic confounders (age, sex and education) on task feasibility will be evaluated.

Reliability will be measured by examining the stability of equivalent tasks completed over the assessment period. This will include examining reliability across parallel test variants that vary by assessment day and by setting (baseline vs remote follow-up).

Practice and learning effects of parallel test variants over repeated assessment will be characterised with linear mixed effects models. Correlational analysis will be completed between parallel test variants, and intraclass correlations will be carried out to examine test–retest reliability of the same test variants administered across days and between settings (virtual study visit vs remote assessment). Coefficients of individual agreement will be used as a measure of interparticipant and intraparticipant variability.

Usability will be examined via the scores on the usability questionnaire for each disease cohort and their matched controls to examine disease cohort-specific problems or difficulties in completing remote assessments. Adherence will be defined as the proportion of participants engaging in daily remote assessments in each indication. Data will be analysed using t-tests, analysis of variance or non-parametric equivalents, as appropriate. Age and education differences between groups will be controlled for as appropriate.

For the end points which require training classifiers or regressors to make predictions, the entire data set will be used for both training machine learning models and validating their statistical properties. Machine learning methods used may include supervised, unsupervised, semi-supervised and reinforcement learning methods for forming intermediate representations of the data and performing the final classification/regression analyses. Speech measures will be derived using several methods from the fields of signal/speech processing and natural language processing to investigate which speech markers are most predictive of clinical status. Some of the planned methods for analyses are described here, but in this broad rapidly evolving field, others may be used where suitable and as they become available.

The sample rate, bit rate and codec compression of smartphones along with speech intelligibility measures will be used to probe dependence of speech features on audio quality measures. Data from the mPower study suggest that phonation data from a bring-your-own-device setup is sufficiently consistent to predict motor disorders to some degree.[83]

Acoustic measures will be extracted via Surfboard, an automated audio feature extraction library,[83] which extracts a range of acoustic speech features (such as mel-frequency cepstrum coefficients, F0 contour, formant frequencies, intensity, loudness); and Vector-Quantized Prosody (VQP), a self-supervised contrastive model for non-timbral prosody.[84] Linguistic and pause analyses will be completed after automated transcription for onward textual analysis. Text features (including number of words, noun and pronoun rate, idea density) will be extracted using Stanza.[85] Speaking time, articulation rate and pause information will be extracted directly from the number of words and timestamps in the transcriptions. Group prediction from text and audio features will be evaluated within the python machine learning package scikit learn[86] with k-fold cross-validation. For ASRTs,

textual analyses will be completed with ParaBLEU, a paraphrase evaluation model,[87] and G-match, evaluating the cosine similarity of textual embeddings between two texts.[45]

Deep learning methods will be used, where the speech measures are derived as internal representations of the models. Methods used here will include unsupervised context-aware word embeddings, a way of distinguishing between identical words with different contextual meanings,[87] and audio representation learning.[84]

Training and validation of the ML models will be performed using cross-validation techniques. Performance of classification analysis will be characterised by the area under the receiver operating characteristic curve (AUC), sensitivity, specificity and Cohen's kappa. Demonstration that a binary classifier performs better than the random baseline will be done via mapping AUC estimates to a p value via the Mann-Whitney U statistic.[88] Results will be contrasted with demographic comparisons (comprising sex, education and age) as in prior analyses,[48] to evaluate the contribution of demographic imbalances.

### Methodological issues and limitations

The primary potential limitation of the study is poor compliance with the self-administered speech tasks over the follow-up period. Efforts to address this potential issue include reminder notifications from the smartphone application and allowing participants to complete these self-assessments at flexible timepoints. It should also be noted that adherence to the self-administered tasks is itself an outcome measure of the study, and so missing data in this regard are informative.

Practice effects on repeated tasks are likely to have an impact on task performance. Parallel test variants, such as those for the ASRT, can minimise practice effects on repeated exposure. However, even with parallel test variants, repeated administration is likely to see task improvements over time related to repeated exposure to the same task and greater familiarity with the test structure and method.[45] For some conditions and some tasks such as category fluency, the practice effects themselves (or lack thereof) may be of interest for identifying specific diagnostic groups, such as individuals with MCI on tasks such as category fluency.[89 90]

Other potential limitations include the fact that recruitment is limited by the requirement for participants to be able to access and use electronic devices for speech tasks and follow-up questionnaires, as not all patients own or have access to connected digital platforms,[40] even with assistance from others.

### Data management and oversight

All aspects of the study will be overseen by the Study Management Group. Primary investigators will oversee and manage all aspects of their study in their respective cohorts. Their responsibility includes monitoring adverse events in their respective cohorts and managing participant discontinuation.

Audio data will be recorded by the app and by the video conference software. On completion of each assessment, the audio will be securely transferred to Novoic's cloud servers, which are fully secure and compliant under relevant security standards (including ISO27001) and privacy regulations (including General Data Protection Regulation). Other clinical data will be stored in a secure password-protected server, compliant with all applicable laws and regulations, including ICH E6 Good Clinical Practice, EU Annex 11, GDPR.

## Data access

The speech data obtained during the study will not be made available to a repository due to the potential for participant identification.

## Patient and public involvement

We have involved patients and members of the public in planning of this study, having sought service users' perspectives on protocol documentation which resulted in amendments to language used and patient facing documentation. We have also implemented changes based on carers' feedback from a focus group of 10–15 individuals. We also actively collect usability feedback from service users. During dissemination, we will invite service users and carers to contribute to a public perspective on the interpretation of trial findings. Results will be made available to participants on request.

## Ethics and dissemination

The study received REC approval from the London—Queen Square Research Ethics Committee on 15 April 2021, and HRA approval from HRA and Health and Care Research Wales on 23 April 2021 (REC reference: 21/PR/0070). Findings will be published in peer-reviewed journals and presented at conferences focused on neuroscience and psychiatry, and machine learning and natural language processing. Novoic will create a project webpage containing key findings, published papers and open-source software packages. Results will be made available to participants on request.

**Acknowledgements** We would like to thank the NIHR Maudsley Biomedical Research Centre's FAST-R service and Service User Advisory Group.

**Contributors** ERH (ORCID ID: 0000-0001-6985-5646): contributed to the writing of the study protocol, in addition to the writing and editing of the manuscript for journal submission. MM (ORCID ID: 0000-0003-4937-3062): main contributor on designing the study and protocol. Contributed to writing the full study protocol and editing the manuscript for journal submission. CS (ORCID ID: 0000-0001-8692-7787): contributed to the study design, writing and editing of the manuscript for journal submission. RS (ORCID ID: 0000-0002-2984-1124): contributed to the editing of the study protocol and manuscript for journal submission, in addition to study set up activities. RHT (ORCID ID: 0000-0001-6742-4842): contributed to editing of the manuscript for journal submission. LC (ORCID ID: 0000-0002-9514-364X): contributed to the writing of the study protocol, in addition to the editing of the manuscript for journal submission. DA (ORCID ID: 0000-0001-6314-216X): contributed to study and protocol design. Principal investigator for neurodegenerative disease cohort. AA-C (ORCID ID: 0000-0002-4924-7712): contributed to study and protocol design. Principal investigator for motor disorders cohort. RC (ORCID ID: 0000-0003-2815-0505): contributed to study and protocol design. Principal investigator for motor disorders cohort. JW (ORCID ID: 0000-0001-5344-7840): contributed to the writing of the study protocol, in addition to editing of the manuscript for journal submission. EF (ORCID ID: 0000-0002-9590-7275): main contributor on designing the study and protocol. Contributed to writing the full study protocol and editing the manuscript for journal submission. AP (ORCID ID: 0000-0002-0805-3356): contributed to editing of the manuscript for journal submission. OA (ORCID ID: 0000-0002-6814-3097): contributed to editing of the manuscript for journal submission. AY (ORCID ID: 0000-0003-2291-6952): chief investigator. Contributed to study and protocol design and edited the manuscript for journal submission.

**Funding** This report is independent research funded by the National Institute for Health Research (Artificial Intelligence, Project RHAPSODY: investigating the clinical feasibility of using AI-based deep audio and language processing techniques to diagnose neurological and psychiatric diseases, AI_AWARD01984) and NHSX.

**Disclaimer** The views expressed in this publication are those of the author(s) and not necessarily those of the National Institute for Health Research, NHSX or the Department of Health and Social Care.

**Competing interests** RS has received an honorarium for speaking from Lundbeck. In the past 3 years, AY has received honoraria for speaking from AstraZeneca, Lundbeck, Eli Lilly and Sunovion; honoraria for consulting from Allergan, Livanova and Lundbeck, Sunovion and Janssen and research grant support from Janssen. ERH declares no conflicts of interest. EF is CEO of Novoic. MM, JW and CS are employees of Novoic, and EF, MM and JW are shareholders in the company.

**Patient and public involvement** Patients and/or the public were involved in the design, or conduct, or reporting, or dissemination plans of this research. Refer to the Methods section for further details.

**Patient consent for publication** Not applicable.

**Provenance and peer review** Not commissioned; externally peer reviewed.

**Data availability statement** No data are available.

**ORCID iDs**
Elliot Hampsey http://orcid.org/0000-0001-6985-5646
Ammar Al-Chalabi http://orcid.org/0000-0002-4924-7712

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
