## [Reviewer comments · BMJ Open]

ARTICLE DETAILS

TITLE (PROVISIONAL)	Protocol for RHAPSODY: a longitudinal observational study examining the feasibility of speech phenotyping for remote assessment of neurodegenerative and psychiatric disorders
AUTHORS	Hampsey, Elliot; Meszaros, Marton; Skirrow, Caroline; Strawbridge, Rebecca; Taylor, Rosie H; Chok, Lazarus; Aarsland, D; Al-Chalabi, Ammar; Chaudhuri, Ray; Weston, Jack; Fristed, Emil; Podlewska, Aleksandra; Awogbemila, Olabisi; Young, Allan

VERSION 1 – REVIEW

REVIEWER	Marrero Aguiar, Victoria Universidad Nacional de Educación a Distancia
REVIEW RETURNED	06-Mar-2022

GENERAL COMMENTS	The attached document contains these suggestions in a more readable format. 1. Is the research question or study objective clearly defined? Yes, but... The long-term goal of this work is of the utmost interest and relevance: to achieve a system that allows discriminating different types of neurodegenerative and psychiatric disorders from speech samples recorded in a mobile application using self-administered speech tasks. But such a goal is presented only as an additional objective in this protocol paper, the main purpose of which is a previous step, that is, to assess the feasibility of this remote collection of continuous narrative. Although this main objective is clearly defined, the introduction does not detail the difficulties derived from it, nor does it address previous work that has faced the corresponding challenge. Instead of reviewing the problems posed by the remote collection of voice and speech recordings, and their use for diagnostic purposes, the paper details healthcare costs of neurological and psychiatric disorders, a topic unrelated to the presentation of the protocol. The challenges associated with the early diagnosis of PNDs are briefly presented, but it would be convenient to review the precedents that have used this same methodology and tried to solve them, e.g. Omberg, et al 2021, De Marchi, et al. 2021, Codina-Filbà, et al. 2021, among others (see below). 2. Is the abstract accurate, balanced and complete? Yes, but if the suggestions of this review are considered, it should be expanded with information regarding the parameters analysed, and the main objectives of the different tasks.
---

	3. Is the study design appropriate to answer the research question? Yes 4. Are the methods described sufficiently to allow the study to be repeated? No First of all, a determining element for the methodology of the study, artificial intelligence, is mentioned in the introduction, ("Novel strategies to improve illness recognition. Artificial intelligence-based techniques could potentially detect subtle and early changes in speech and translate them into predictive diagnostic models", p. 4, line 16). But it does not reappear in the rest of the paper. In addition, "the study will examine whether acoustic and linguistic patterns can be used to distinguish from group-specific control participants, and from other clinical indications, and whether this is impacted by relevant contextual or disease information covariates" (p. 4, l. 44-47). So, to allow the study to be repeated, it would be necessary a brief overview of the AI system that will make it possible to analyse the data. (a) As regards the acoustic data, the speech recognition algorithms used, whether recurrent or deep neural network models have been employed, the parameters that have been analysed, both in the frequency and time domain. b) For linguistic data, the system of clinical natural language processing (NLP) system used: MedLEE (Friedman 1997)? cTAKES (Savova et al. 2010)? MetaMap or MetaMap Lite (Demner-Fushman, Rogers & Aronson 2017)? CLAMP (Soysal et al. 2018)? None of them? In this case, where is the system described? It should be stated the part-of-speech tagger (and if retrained on a clinical corpus), the sentence detector system, the tokenizer, the name entity recognizer (does it uses the Unified Medical Language System?), the rule engine, etc. In addition, to ensure reproducibility, it would be necessary to define more clearly the expected results of the tasks and the parameters that are measured in each one (see point 6). 5. Are research ethics (e.g. participant consent, ethics approval) addressed appropriately? Yes 6. Are the outcomes clearly defined? No The main objectives of the analysis are, apart from usability:  - Feasibility, measured as the length of significant speech generated during elicitation tasks (p. 9, l. 50). There is no indication of how it will be decided whether the speech produced is meaningful or not. Will the proportion of empty pauses, of unanalysable elements, be taken into account? Will a criterion of minimum grammatical structure be established? - Reliability across parallel task variants. Parallel task variants are not clearly defined (The Automated Story Recall Task with immediate recall is parallel to The Automated Story Recall Task with delayed recall, for instance?), But above all, how will the learning effect be monitored? Phonemic and semantic tasks, for example, will be performed five times in 28 days; even if letters or semantic categories are changed from one time to the next (something that is not defined in the article, and would require a control of variables as frequency of use to guarantee their equivalence), the procedure will become more familiar over time, especially to a population without memory problems, such as the affective disorders group. The same applies to the Automated Story Recall Task: will be different the 12 times that these two
--	---

	tasks will be performed? Will there be 24 stories with exactly equivalent characteristics (length, prosodic patterns, lexical frequency, grammatical complexity, argumentative structure, etc.)? Or will there be only one or two stories which the subjects will have to repeat twelve times in 28 days? In the latter case, it is obvious that the variants will not be comparable, because in the end they will know the story by heart. It is stated that "the stories are matched for several linguistic and phonological qualities" (p. 9, l. 28-29). What are these linguistic and phonological qualities? Has the effect of sociocultural level been controlled for (Kawano, et al 2010)? In addition to these outcomes linked to the three objectives listed above, it would be desirable to justify, at least minimally, the use of this sequence of 12 tasks (table 2), and what is expected to be obtained from each of them in the three groups of patients. None of them is strictly novel, they are generally well known and widely used in cognitive explorations. But they have different characteristics, goals, and constraints.  1. Story Recall is a classic task to assess verbal episodic memory. It makes perfect sense and has often been used to assess the onset of diseases such as Alzheimer's, but what is the expected outcome in patients that do not have memory impairment, such as those with affective disorders? And in Parkinson's or motor neuron patients, are different outcomes expected than in Alzheimer's? 2. Picture description tasks are often used to assess visual, verbal or semantic memory, but both its graphical characteristics and the way of measuring the response have been carefully calibrated (for instance, the Family Pictures of WAIS-III, or the "cookie left" in the Boston Aphasia Test; see Mueller, Hermann, Mecollari & Turkstra 2018 for a review). This protocol paper refers that a "singular image" (p.7, l. 49) will be described. Will some of these already well-known images be used, or is this a new one? If so, by what criteria has it been created, what is it intended to measure, and how will the responses be assessed? Something similar applies to the sequence narration task: "Participants must describe their narrative interpretation of a series of images presented as a storyboard." (p.7, l. 52-53). have the variables that determine the complexity of this task been controlled for? 3. Fluency tasks aim to analyse lexical access, either by the phonological route ("letter" fluency tasks) or by the semantic one (nouns in the "semantic" fluency task or verbs in the "action" fluency task). What is expected of these three tasks in each clinical group? Poorer semantic than phonemic fluency is often reported in Alzheimer's Disease, but the meta-analysis of Laws, Duncan & Gale (2010) found this profile also in healthy elderly subjects; will the age variable be considered and how will it be controlled? It has also been found that the differences between action and naming fluency tasks may be useful for differentiating Primary Progressive Aphasias, PPA (Costa Beber & Chaves 2014); is this task expected to produce different results among patients in Group 1: Neurodegenerative cognitive disorders (Alzheimer's Disease versus Lewi Bodies versus PPA)? 4. Counting 1-10 is a classic digit span task associated with attention and executive function processes (Hale, Hoepfner & Fiorello, 2002), or 10 to 1 (digit span backwards, working memory: Hilbert et al. 2014). Shroeder et al. (2012) recommend using it with caution in populations with severe memory disorders. This task is included in table 3, but not in table 2. The same applies to stroop (that has not been described or justified).
--	---

	5. Acoustic features are supposed to be measured in tasks such as sustained phonation, syllable repetition, perhaps even sentence repetition, or reading a script. But again, this is not clearly indicated in the paper. Sustained phonation is intended to assess the efficiency of glottal functioning: is this interesting in all groups? But if it is not known which syllables are to be repeated, which sentences, with what criteria they are constructed, and the characteristics of the script to be read, it is difficult to estimate the goals and results of these tasks. On the other hand, what kind of acoustic features will be taken into account? In glottal features (jitter, shimmer, harmonic-to-noise ratio, etc.), how will interference, noise, codec compression of smartphones be controlled? Prosodic features as f0 range, speech rate, pause duration, etc., depends directly on the characteristics of the stimulus. And finally, reading introduces a new variable: this is a skill very different from those related to the rest of the language tasks, and again, dependent on the level of schooling. In short, although this point of the review may seem long, it can be summarised in a recommendation that can be incorporated into the work in a synthetic form: to expand the information in table 2, including a more precise description of the tasks (which should coincide with those in table 3), and adding two columns, one for expected outcomes and other for limitations of each task. 7. If statistics are used are they appropriate and described fully? Not entirely The above remarks on outcomes (question 6) would have statistical implications. But it would also be useful to clarify some other details:  - Speech measures: what methods will be used to discriminate the most predictive markers? The expression "several methods from the fields of signal/speech processing and natural language processing" (pg. 10, lines 21-22) is somewhat imprecise. - What kind of "audio representation" (pg. 10, line 26) will be used? 8. Are the references up-to-date and appropriate? Not entirely Some suggestions:  - On artificial intelligence-based /machine learning / big data approaches to detect changes in the speech of this clinical population:  • Shatte, A. B., Hutchinson, D. M., & Teague, S. J. (2019). Machine learning in mental health: a scoping review of methods and applications. Psychological medicine, 49(9), 1426-1448. • Patel, U. K., Anwar, A., Saleem, S., Malik, P., Rasul, B., Patel, K., ... & Arumaithurai, K. (2021). Artificial intelligence as an emerging technology in the current care of neurological disorders. Journal of neurology, 268(5), 1623-1642. • Li, R., Wang, X., Lawler, K., Garg, S., Bai, Q., & Alty, J. (2022). Applications of Artificial Intelligence to aid detection of dementia: a scoping review on current capabilities and future directions. Journal of Biomedical Informatics, 104030. • Garrard P, Rentoumi V, Gesierich B, Miller B, Gorno-Tempini ML (2014) Machine learning approaches to diagnosis and laterality effects in semantic dementia discourse. Cortex 55, 122-129. • Lehr, M., Prud'hommeaux, E., Shafran, I., & Roark, B. (2012). Fully automated neuropsychological assessment for detecting mild cognitive impairment. In Thirteenth Annual Conference of the
--	--

	International Speech Communication Association. [They use the technique of retelling stories, as in this study.] - On assessment of spontaneous /narrative /connected speech in this clinical population:  • Fasih Haider, Sofia De La Fuente, Saturnino Luz (2019). An assessment of paralinguistic acoustic features for detection of alzheimer's dementia in spontaneous speech. IEEE J. Select. Top. Signal Process., 14 (2), pp. 272-281. • Bahman Mirheidari, Daniel Blackburn, Traci Walker, Markus Reuber, Heidi Christensen (2019). Dementia detection using automatic analysis of conversations. Computer Speech & Language, 53 (2019), pp. 65-79 • Fraser KC, Meltzer JA, Rudzicz F. Linguistic features identify Alzheimer's disease in narrative speech. J Alzheimers Dis. 2016;49:407–22. • König, A., Satt, A., Sorin, A., Hoory, R., Toledo-Ronen, O., Derreumaux, A., ... & David, R. (2015). Automatic speech analysis for the assessment of patients with predementia and Alzheimer's disease. Alzheimer's & Dementia: Diagnosis, Assessment & Disease Monitoring, 1(1), 112-124. • Mueller, K. D., Hermann, B., Mecollari, J., & Turkstra, L. S. (2018). Connected speech and language in mild cognitive impairment and Alzheimer's disease: A review of picture description tasks. Journal of clinical and experimental neuropsychology, 40(9), 917-939. - On acoustic or linguistic measures to early detection of neurological diseases:  • Meilán, J. J. G., Martínez-Sánchez, F., Carro, J., López, D. E., Millian-Morell, L., & Arana, J. M. (2014). Speech in Alzheimer's disease: can temporal and acoustic parameters discriminate dementia? Dementia and geriatric cognitive disorders, 37(5-6), 327-334. • Poole, M. L., Brodtmann, A., Darby, D., & Vogel, A. P. (2017). Motor speech phenotypes of frontotemporal dementia, primary progressive aphasia, and progressive apraxia of speech. Journal of Speech, Language, and Hearing Research, 60(4), 897-911. • Cummins, N., Scherer, S., Krajewski, J., Schnieder, S., Epps, J., & Quatieri, T. F. (2015). A review of depression and suicide risk assessment using speech analysis. Speech communication, 71, 10-49. 9. Do the results address the research question or objective? N/A 10. Are they presented clearly? N/A 11. Are the discussion and conclusions justified by the results. N/A 12. Are the study limitations discussed adequately? No The only paragraph discussing one of the limitations of the study, poor compliance, appears on p.10, l. 36-42, where some measures to address this potential issue are also mentioned. Apart from this, the only acknowledged limitations are listed in pg. 2, along with the strenghts: "study design uniquely allows for investigation of the same tasks across a wide number of indications", "Limited sample size" and "Short follow-up period". However, as noted in point 6, there are many other potential limitations arising from, at least, the following factors:
--	---

	a) The type of stimuli used: effects of lexical frequency, grammatical complexity, argumentative structure, etc. b) Effects of the channel used: telephone or videoconferencing platform. c) Effects of learning by repetition of tasks. d) Effects derived from the characteristics of the sample, which may be different in the neurodegenerative and psychiatric disorder groups. 13. Is the supplementary reporting complete (e.g. trial registration; funding details; CONSORT, STROBE or PRISMA checklist)? Yes, but... The instructions for reviewers of study protocols state that "Protocol papers should report planned or ongoing studies. The dates of the study should be included in the manuscript." This study clearly mentions that participants completed online questionnaires in a total of 12 days spread over four weeks, but I have not found the specific dates on which these tasks were carried out. 14. To the best of your knowledge is the paper free from concerns over publication ethics (e.g. plagiarism, redundant publication, undeclared conflicts of interest)? Yes 15. Is the standard of written English acceptable for publication? Yes References mentioned Codina-Filbà, J., Escalera, S., Escudero, J., Antens, C., Buch-Cardona, P., & Farrús, M. (2021, June). Mobile eHealth platform for home monitoring of bipolar disorder. In International Conference on Multimedia Modeling (pp. 330-341). Springer, Cham. Costa Beber, B. & Chaves, M. L. (2014). The basis and applications of the action fluency and action naming tasks. Dementia & Neuropsychologia, 8, 47-57. De Marchi, F., Contaldi, E., Magistrelli, L., Cantello, R., Comi, C., & Mazzini, L. (2021). Telehealth in neurodegenerative diseases: opportunities and challenges for patients and physicians. Brain Sciences, 11(2), 237. Demner-Fushman, D., Rogers, W. J., & Aronson, A. R. (2017). MetaMap Lite: an evaluation of a new Java implementation of MetaMap. Journal of the American Medical Informatics Association, 24(4), 841-844. Friedman, C. (1997). Towards a comprehensive medical language processing system: methods and issues. In Proceedings of the AMIA annual fall symposium (p. 595). American Medical Informatics Association. Hale, J. B., Hoepfner, J. A. B., & Fiorello, C. A. (2002). Analyzing digit span components for assessment of attention processes. Journal of psychoeducational assessment, 20(2), 128-143. Hilbert, S., Nakagawa, T. T., Puci, P., Zech, A., & Bühner, M. (2014). The digit span backwards task. European Journal of Psychological Assessment. Kawano, N., Umegaki, H., Suzuki, Y., Yamamoto, S., Mogi, N., & Iguchi, A. (2010). Effects of educational background on verbal fluency task performance in older adults with Alzheimer's disease
--	---

	and mild cognitive impairment. International Psychogeriatrics, 22(6), 995-1002. Laws, K. R., Duncan, A., & Gale, T. M. (2010). 'Normal'semantic-phonemic fluency discrepancy in Alzheimer's disease? A meta-analytic study. Cortex, 46(5), 595-601. Omberg, L., Chaibub Neto, E., Perumal, T. M., Pratap, A., Tediarjo, A., Adams, J., ... & Mangravite, L. M. (2021). Remote smartphone monitoring of Parkinson's disease and individual response to therapy. Nature Biotechnology, 1-8. Savova, G. K., Masanz, J. J., Ogren, P. V., Zheng, J., Sohn, S., Kipper-Schuler, K. C., & Chute, C. G. (2010). Mayo clinical Text Analysis and Knowledge Extraction System (cTAKES): architecture, component evaluation and applications. Journal of the American Medical Informatics Association, 17(5), 507-513. Schroeder, R. W., Twumasi-Ankrah, P., Baade, L. E., & Marshall, P. S. (2012). Reliable digit span: A systematic review and cross-validation study. Assessment, 19(1), 21-30. Soysal, E., Wang, J., Jiang, M., Wu, Y., Pakhomov, S., Liu, H., & Xu, H. (2018). CLAMP—a toolkit for efficiently building customized clinical natural language processing pipelines. Journal of the American Medical Informatics Association, 25(3), 331-336.
--	---

REVIEWER	König, Alexandra Institut National de Recherche en Informatique et en Automatique Centre de Recherche Sophia Antipolis Méditerranée
REVIEW RETURNED	12-Mar-2022

GENERAL COMMENTS	Overall, the paper describes the protocol for a longitudinal study which investigates the use of speech phenotyping for remote assessment of neurodegenerative and psychiatric disorders. The study aims to collect narrative speech data via an app of subjects of three different patient cohorts. Primary outcome is feasibility of the method, and secondary is if certain speech patterns help to predict diagnostic groups. The study seems overall highly relevant to the current field of finding early diagnostic markers for cognition and affective disorders but also very ambitious which could represent a certain risk for success. It seems that many aspects are combined into one protocol; evaluating feasibility, predicting diagnosis through speech and help with differential diagnosis thus distinguishing between different types of diseases which all together could potentially be difficult to achieve in one study. It would be interesting to get more information about the three different patient cohorts, their names, design, baseline assessments, if there are still ongoing, etc. This is interesting to know since harmonizing between all these cohorts and patient groups may be quite challenging. Another important question is how are these additional speech tasks not intervening with the ongoing cohorts and their regular assessments? Probably there need to be parallel versions administered in order to avoid learning effects? This should be clarified more. The 'affective disorder' group consists only of patients between 18-65 compared to the others 50-85, this could cause a significant difference in age between the groups. How this will be handled? Also patients only which diagnosed with Major Depression seem to be included. Thus, why not simply call it then 'Depression group'?
--

otherwise it could be easily assumed that other affective disorders would be included as well in this group.

As a potential limitation it could be mentioned that this study only allows subjects to participate who have access to a smartphone, internet connection and are familiar with using such devices. This is not always evident for certain elderly people.

Further comments:

Introduction

- First paragraph : up from line 6, the sentence starting with 'Herewith,...' describes the aim of the study, which is placed usually more at the end of Intro section
- Page 4, line 10: 'Failure to intervene...' This sentence should be rephrased, it sounds not correct 'the impact ON the factors EFFECTING (is it not AFFECTING?)'

Methods:

- Error Figure 1. Is probably suppose to be Table 1. In this table 'Speech and Language assessments' are indicated, but it would be good to specify which ones
- A major concern is that not all patient groups receive the same baseline assessment, particularly one for cognition. Why only group 1 receives the TICS-M and not the others? Same for a brief depression screening, it would be important to administer similar tests, scales to all participants in order to harmonize and compare patient groups among each other.
- Page 7, up from line 9, the description of the procedure is not really clear and a bit confusing...is screening = main assessment? Are there two separate steps?
- The speech task protocol (table 2.) seems relatively long this could be for certain patients quite tiring. What will be the approximate amount of time to perform all these tasks? Include this information please. Maybe consider less tasks?
- Page 8, up from line 39, should visual and hearing impairment not be checked right at the beginning event before baseline, maybe as a sort of exclusion criteria? Seems a bit late to do that after the first assessments.
- Table 1. is in the paper presented after table 2.
- Table 2., regarding certain tests, how will be controlled for learning effects? Parallel versions? In particular for the 'Automated Story Recall task'?
- Page 9, line 38: it is not only 'continuous narrative speech' but also very recordings of very directed short cognitive tasks
- Line 47, Why as a primary endpoint only considering participation of the main assessment?
- Page 10, up from line 21: A bit vague to indicate only the term 'speech markers', it would be helpful to describe at least the different types of speech measures that will be extracted for this study.

	- Up from line 26: How are the beforementioned linguistic features extracted and patterns analyzed ? - Up from line 53: Is Novoic's secure server certified to host health data? Speech data in this context falls into personal health data and thus it is not possible to anonymize, therefore security measures for this data have to be well documented. The 'secure password-protected server' is located where? At the hospital? University? This needs to be clarified. - Additional comment, 5 authors of the manuscript are employees of the company Novoic which represents a strong involvement of an industry partner in the design of the clinical study;
--	---

VERSION 1 – AUTHOR RESPONSE

Reviewer 1

We would like to thank reviewer 1 for their considered review of our manuscript. We appreciate the opportunity to have provided further clarification throughout the text which we feel should aid reader understanding of the study process.

It would be interesting to get more information about the three different patient cohorts, their names, design, baseline assessments, if there are still ongoing, etc. This is interesting to know since harmonizing between all these cohorts and patient groups may be quite challenging.

Details of the four groups are presented under 'participants'. Study recruitment is ongoing – specific dates have been added (also under 'participants'). Tables 1 and 2 have been updated to outline, in detail, what assessments each group will undergo.

Another important question is how are these additional speech tasks not intervening with the ongoing cohorts and their regular assessments? Probably there need to be parallel versions administered in order to avoid learning effects? This should be clarified more.

In table 2, we now clarify that ASRT stories are available in multiple (N=36 parallel) variants, and provide a reference where story task metrics are documented and evaluated (Skirrow et al., 2022).

We clarify that the stories are matched for presentation rate, and the number of words, number of sentences, number of dependent clauses, mean sentence length, and ratio of dependent clauses to t-units.

The referenced paper (Skirrow et al., 2022) also shows good parallel forms reliability of the ASRT tasks, and limited improvements over time when administered daily to older adults with mild cognitive impairment or mild Alzheimer's disease, and older adults that are cognitively unimpaired. These practice effects can be readily evaluated in longitudinal mixed model analyses, where linear trends over time can be incorporated.

The 'affective disorder' group consists only of patients between 18-65 compared to the

others 50-85, this could cause a significant difference in age between the groups. How this will be handled?

We thank the reviewer also for pointing out this issue, which was raised by a researcher on our team prior to data collection and has been addressed via a project amendment. We have now updated the age range described.

However, the reviewer is correct to note there will likely be age differences between the groups, not only due to participant selection criteria (which parallel age ranges for onset and incidence), but also due to natural disease progression and the ages where these diseases naturally occur. We now provide more information in the 'Statistical analysis plan' section, describing how age and broader demographic differences will be controlled for or evaluated in analyses.

Also patients only which diagnosed with Major Depression seem to be included. Thus, why not simply call it then 'Depression group'? otherwise it could be easily assumed that other affective disorders would be included as well in this group.

We think there may be a slight misunderstanding here – participants in group 3 must be currently in a depressive episode, but this can be in the context of bipolar disorder as well as major depression, as stated in the final line of the paragraph in the 'participants' section that begins "Group 3:". This section has been reordered slightly to provide clarity to the reader.

As a potential limitation it could be mentioned that this study only allows subjects to participate who have access to a smartphone, internet connection and are familiar with using such devices. This is not always evident for certain elderly people.

We have added this point to the end of the 'Methodological issues and limitations' section.

Further comments:

- First paragraph : up from line 6, the sentence starting with 'Herewith,...' describes the aim of the study, which is placed usually more at the end of Intro section

We have reordered our introduction to reflect this advice.

- Page 4, line 10: 'Failure to intervene...' This sentence should be rephrased, it sounds not correct 'the impact ON the factors EFFECTING (is it not AFFECTING?)'

We thank you for pointing out this error, which has now been corrected to read 'affecting'.

Methods:

- Error Figure 1. Is probably suppose to be Table 1. In this table 'Speech and Language assessments' are indicated, but it would be good to specify which ones

We are slightly unclear regarding this feedback, as figure 1 and table 1 are separate items. Figure 1 is a flow-diagram of participant journey in the study, whereas table 1 outlines screening tools and questionnaires used. Journal guidelines required us to remove the figures from the main document and add them in a separate file. Has this reviewer been able to review the file 'rhapsody protocol figure 1.pdf' which we uploaded on submission, as this may clarify the difference between figure 1 and table 1.

The text for 'speech and language assessments' now includes '(see table 2)' to outline the rundown of the speech tasks used

- A major concern is that not all patient groups receive the same baseline assessment, particularly one for cognition. Why only group 1 receives the TICS-M and not the others? Same for a brief depression screening, it would be important to administer similar tests, scales to all participants in order to harmonize and compare patient groups among each other.

Different scales are administered to different groups to adequately clinically characterise participants, and their eligibility, within their respective groupings. For example, group 1 receive the TICS-M because a score of 20-30 is required for inclusion. Whilst we appreciate that consistent use of assessments across all groups would be informative, the current assessments are instead targeted to provide the required eligibility information whilst limiting participant burden.

- Page 7, up from line 9, the description of the procedure is not really clear and a bit confusing...is screening = main assessment? Are there two separate steps?

This text has been reworked to provide clarity. In short, the study visit is one teleconference split into two parts. Screening and then (for those eligible) the baseline data collection.

- The speech task protocol (table 2.) seems relatively long this could be for certain patients quite tiring. What will be the approximate amount of time to perform all these tasks? Include this information please. Maybe consider less tasks?

This protocol reports an ongoing study for which we have collected much of the baseline data, meaning the task protocol won't be changed at this stage. Information on the approximate amount of time to complete the tasks has been added

- Page 8, up from line 39, should visual and hearing impairment not be checked right at the beginning event before baseline, maybe as a sort of exclusion criteria? Seems a bit late to do that after the first assessments.

Hearing and visual impairments are done as part of the screening checks as well. Text has been added to reflect this.

- Table 1. is in the paper presented after table 2.

This oversight has now been corrected.

- Table 2., regarding certain tests, how will be controlled for learning effects? Parallel versions? In particular for the 'Automated Story Recall task'?

In table 2, we now clarify which repeated tests have parallel versions, and table 3 provides an overview of where parallel variants are used, and where individual stories or tasks are repeated.

The referenced paper (Skirrow et al., 2022) also shows good parallel forms reliability of the ASRT tasks, and limited improvements over time when administered daily to older adults with mild cognitive impairment or mild Alzheimer's disease, and older adults that are cognitively unimpaired. These practice effects can be readily evaluated in longitudinal mixed model analyses, where linear trends over time can be incorporated. This has now been described explicitly in the 'statistical analysis plan' section.

- Page 9, line 38: it is not only 'continuous narrative speech' but also very recordings of

very directed short cognitive tasks

Thank you for pointing this out. This sentence has been corrected to read 'speech data' as a more inclusive term.

- Line 47, Why as a primary endpoint only considering participation of the main assessment?

This is not the case, and we thank the reviewer for pointing out this error in our writing. The primary endpoint should consider ASRT task assessment feasibility both in the main assessment phase and remotely. We have corrected this in text.

- Page 10, up from line 21: A bit vague to indicate only the term 'speech markers' , it would be helpful to describe at least the different types of speech measures that will be extracted for this study.

We now provide more detail on text and audio features which will form part of our analyses in the 'statistical analysis' plan section.

- Up from line 26: How are the beforementioned linguistic features extracted and patterns analyzed?

In the statistical analysis plan we now provide more detailed information relating to the acoustic and linguistic representations that will be extracted, and how they will be analysed along with relevant references for additional information.

- Up from line 53: Is Novoic's secure server certified to host health data? Speech data in this context falls into personal health data and thus it is not possible to anonymize, therefore security measures for this data have to be well documented. The 'secure password-protected server' is located where? At the hospital? University? This needs to be clarified.

Novoic's cloud servers are fully secure and compliant under relevant security standards (including ISO27001) and privacy regulations (including GDPR). This information is provided in the section entitled 'data management and oversight'.

- Additional comment, 5 authors of the manuscript are employees of the company Novoic which represents a strong involvement of an industry partner in the design of the clinical study;

This is correct, however we include this information in the 'competing interests statement' at the conclusion of the manuscript.

Reviewer 2

1. Is the research question or study objective clearly defined? Yes, but...

The long-term goal of this work is of the utmost interest and relevance: to achieve a system that allows discriminating different types of neurodegenerative and psychiatric disorders from speech samples recorded in a mobile application using self-administered speech tasks. But such a goal is presented only as an additional objective in this protocol paper, the main purpose of which is a previous step, that is, to assess the feasibility of this remote collection of continuous narrative.

Although this main objective is clearly defined, the introduction does not detail the difficulties derived from it, nor does it address previous work that has faced the corresponding challenge. Instead of reviewing the problems posed by the remote collection of voice and speech recordings, and their use for diagnostic purposes, the paper details healthcare costs of neurological and psychiatric disorders, a topic unrelated to the presentation of the protocol. The challenges associated with the early diagnosis of PNDs are briefly presented, but it would be convenient to review the precedents that have used this same methodology and tried to solve them, e.g. Omberg, et al 2021, De Marchi, et al. 2021, Codina-Filbà, et al. 2021, among others (see below).

We now include a novel section in the introduction, entitled 'digital and remote assessment strategies' to described digital and higher-frequency remote assessments, and advantages and challenges associated with this form of assessment.

2. Is the abstract accurate, balanced and complete?

Yes, but if the suggestions of this review are considered, it should be expanded with information regarding the parameters analysed, and the main objectives of the different tasks.

Although somewhat limited by abstract word count, we have updated the abstract to include information on some of the parameters to be analysed, and details on the objectives of the speech tasks.

3. Is the study design appropriate to answer the research question? Yes

4. Are the methods described sufficiently to allow the study to be repeated? No

First of all, a determining element for the methodology of the study, artificial intelligence, is mentioned in the introduction ... to allow the study to be repeated, it would be necessary a brief overview of the AI system that will make it possible to analyse the data.

(a) As regards the acoustic data, the speech recognition algorithms used, whether recurrent or deep neural network models have been employed, the parameters that have been analysed, both in the frequency and time domain.

AI-assisted analyses are now presented in more detail in the 'Statistical analysis plan' section, describing both the acoustic and linguistic feature extraction approach, and textual and prosodic analysis methods that will be used. However, we also note in this section that in this rapidly growing field, other methods may be adopted after study completion, should they be suitable.

For linguistic data, the system of clinical natural language processing (NLP) system used: MedLEE (Friedman 1997)? cTAKES (Savova et al. 2010)? MetaMap or MetaMap Lite (Demner-Fushman, Rogers & Aronson 2017)? CLAMP (Soysal et al. 2018)? None of them? In this case, where is the system described? It should be stated the part-of-speech tagger (and if retrained on a clinical corpus), the sentence detector system, the tokenizer, the name entity recognizer (does it uses the Unified Medical Language System?), the rule engine, etc.

In addition, to ensure reproducibility, it would be necessary to define more clearly the expected results of the tasks and the parameters that are measured in each one (see point 6).

Text features (including, number of words, noun, pronoun and verb rate, idea density) will be extracted using Stanza, a python package for natural language analysis (Qi et al., 2020). Speaking time, articulation rate, and pause information will be extracted directly from the number of words and timestamps in the transcriptions.

5. Are research ethics (e.g. participant consent, ethics approval) addressed appropriately? Yes

6. Are the outcomes clearly defined? No

The main objectives of the analysis are, apart from usability:

- Feasibility, measured as the length of **significant** speech generated during elicitation tasks (p. 9, l. 50). There is no indication of how it will be decided whether the speech produced is meaningful or not. Will the proportion of empty pauses, of unanalysable elements, be taken into account? Will a criterion of minimum grammatical structure be established?

As the reviewer points out, the primary objective is to evaluate feasibility of these kinds of assessments within the targeted indications. They correctly point out that there is much more to speech than the duration of vocalisation, which is the primary, but not only metric, which will be evaluated. We now provide additional information in the section 'statistical analysis plan' where we define the acoustic and textual measures (including empty pauses) that will be evaluated.

- Reliability across parallel task variants.

ASRT task:

Parallel task variants are not clearly defined (The Automated Story Recall Task with immediate recall is parallel to The Automated Story Recall Task with delayed recall, for instance?), But above all, how will the learning effect be monitored? T... will be different the 12 times that these two tasks will be performed? ...What are these linguistic and phonological qualities?

The reviewer raises important concerns around practice and familiarity effects, particularly for the Automatic Story Recall Task (ASRT) which will be administered twice daily

In table 2, we now clarify that ASRT stories are available in multiple (N=36 parallel) variants, and provide a reference where story task metrics are documented and evaluated (Skirrow et al., 2022).

We clarify that the stories are matched for presentation rate, and the number of words, number of sentences, number of dependent clauses, mean sentence length, and ratio of dependent clauses to t-units.

The referenced paper (Skirrow et al., 2022) also shows good parallel forms reliability of the ASRT tasks, and limited improvements over time when administered daily to older adults with mild cognitive impairment or mild Alzheimer's disease, and older adults that are cognitively unimpaired. These practice effects can be readily evaluated in longitudinal mixed model analyses, where linear trends over time can be incorporated. This has now been described explicitly in the 'statistical analysis plan' section.

Phonemic and semantic tasks, for example, will be performed five times in 28 days; even if letters or semantic categories are changed from one time to the next (something that is not defined in the article, and would require a control of variables as frequency of use to guarantee their equivalence), the procedure will become more familiar over time, especially to a population without memory problems, such as the affective disorders group.

In table 2, we now also specify the different prompts used for semantic fluency and letter fluency tasks.

We agree with the reviewer, that even with parallel forms of the same test we would likely see task improvements over time related to repeated exposure to the same task and greater familiarity with the test structure and method. This point is now included in the study limitations where we have included a paragraph to discuss practice effects.

Has the effect of sociocultural level been controlled for(Kawano, et al 2010)?

Control groups will be recruited matched to each group for sex, age and education levels. For case-control comparisons, therefore, this will allow for the effects of education to be controlled for -which were found to be important in Kawano et al., 2010. We now specify that these demographic effects will be covaried for, or evaluated in the 'statistical analysis plan' section.

In addition to these outcomes linked to the three objectives listed above, it would be desirable to justify, at least minimally, the use of this sequence of 12 tasks (table 2), and what is expected to be obtained from each of them in the three groups of patients.....In short, although this point of the review may seem long, it can be summarised in a recommendation that can be incorporated into the work in a synthetic form: to expand the information in table 2, including a more precise description of the tasks (which should coincide with those in table 3), and adding two columns, one for expected outcomes and other for limitations of each task.

We thank the reviewer for highlighting these limitations in our task reporting, and for the thorough and constructive recommendations. As recommended by the reviewer we have extended table 2 to include an overview of task characteristics and evidence from prior research in the evaluated indications. This includes a separate titled sections 'verbal cognitive and speech tasks'. Here we also describe rationale for the selection of the tests, and specify that we expect task performance differences between the clinical indications evaluated. More details are provided below in response to specific comments/points, below:

1. Story Recall is a classic task to assess verbal episodic memory. It makes perfect sense and has often been used to assess the onset of diseases such as Alzheimer's, but what is the expected outcome in patients that do not have memory impairment, such as those with affective disorders? And in Parkinson's or motor neuron patients, are different outcomes expected than in Alzheimer's?
 - In Table 2 column 3, we described prior research showing greater impairment in MCI than in DLB and PD groups in overall performance on this task. To our knowledge there has not yet been a head-to-head investigation of impairments on this task in neurological and neurodegenerative conditions and affective disorders.
 1. Picture description tasks are often used to assess visual, verbal or semantic memory, but both its graphical characteristics and the way of measuring the response have been carefully calibrated (for instance, the Family Pictures of WAIS-III, or the "cookie left" in the Boston Aphasia Test; see Mueller, Hermann, Mecollari & Turkstra 2018 for a review). This protocol paper refers that a "singular image" (p.7, l. 49) will be described. Will some of these already well-known images be used, or is this a new one? If so, by what criteria has it been created, what is it intended to measure, and how will the responses be assessed?
 - We now provide specification of the images that will be used for picture description, alongside relevant references. These are all established and validated picture description tasks.
 1. Something similar applies to the sequence narration task: "Participants must describe their narrative interpretation of a series of images presented as a storyboard." (p.7, l. 52-53). have the variables that determine the complexity of this task been controlled for?
 - The images presented are previously published, and from Nicholas and Brookshire (1993). The reference is now provided.
 1. Fluency tasks aim to analyse lexical access, either by the phonological route ("letter" fluency tasks) or by the semantic one (nouns in the "semantic" fluency task or verbs in the "action" fluency task). What is expected of these three tasks in each clinical group? Poorer semantic than phonemic fluency is often reported in Alzheimer's Disease, but the meta-analysis of Laws, Duncan & Gale (2010) found this profile also in healthy elderly subjects; will the age variable be considered and how will it be controlled? It has also been found that the differences between action and naming fluency tasks may be useful for differentiating Primary

Progressive Aphasias, PPA (Costa Beber & Chaves 2014); is this task expected to produce different results among patients in Group 1: Neurodegenerative cognitive disorders (Alzheimer's Disease versus Lewi Bodies versus PPA)?

- We now describe in table 2 prior literature on task performance in semantic, phonemic and action fluency. The reviewer points out that differences have been found previously within the clinical groups evaluated (between DLB and AD). This is now described within table 2.
- For age-related changes we expect to evaluate and control for these statistically, which has been noted in the 'statistical analysis plan' section of the manuscript.
 1. Counting 1-10 is a classic digit span task associated with attention and executive function processes (Hale, Hoepfner & Fiorello, 2002), or 10 to 1 (digit span backwards, working memory: Hilbert et al. 2014). Shroeder et al. (2012) recommend using it with caution in populations with severe memory disorders.
- Counting 1-10 and 10-1 was used as a brief familiarisation task for participants to get used to the testing interface and interacting with the app. We now clarify this in table 2
- However, the reviewer is providing references for digit span (Hale, Hoepfner & Fiorello, 2002; Hilbert et al., 2014) which was also evaluated in the study, and further clarification regarding test characteristics is now also provided in table 2.
 1. This task [counting 1-10] is included in table 3, but not in table [2]. The same applies to Stroop (that has not been described or justified).
- Table 2 was previously used to provide an overview of tasks administered during the main assessment, and table 3 to provide an overview of tasks administered during remote assessments. In line with the reviewer's recommendations, however, we have amalgamated the two tables, so that table 2 now provides a more thorough overview of speech eliciting tasks that are recorded, and table 3 now provides an overview of scheduled assessments across main (supervised) and remote assessments.
- Table 2 now describes the stroop and counting tasks in greater detail.
 1. Acoustic features are supposed to be measured in tasks such as sustained phonation, syllable repetition, perhaps even sentence repetition, or reading a script. But again, this is not clearly indicated in the paper. Sustained phonation is intended to assess the efficiency of glottal functioning: is this interesting in all groups? But if it is not known which syllables are to be repeated, which sentences, with what criteria they are constructed, and the characteristics of the script to be read, it is difficult to estimate the goals and results of these tasks.
- Further information on sustained phonation, syllable and sentence repetition, and script reading are now provided in table 2. References are provided for the tasks where available.
- Although as pointed out by the reviewer, measures attained during sustained phonation may more informative in some groups, the head-to-head nature of the comparison of speech tasks across diagnostic groups allows task completion and task feasibility to be evaluated overall, alongside more exploratory measures of task performance characteristics.
 1. On the other hand, what kind of acoustic features will be taken into account?
- Acoustic features will be extracted with Novoic's automated audio feature extraction library, surfboard (Lenain et al., 2020); and VQP to extract non-timbral prosody (Weston et al., 2021). This is now described in more detail in the section 'statistical analysis plan'
 1. In glottal features (jitter, shimmer, harmonic-to-noise ratio, etc.), how will interference, noise, codec compression of smartphones be controlled?
- We have described our approach further in the section 'statistical analysis plan' where we specify that the sample rate, bit rate, codec etc along with speech intelligibility measures will be used to probe dependence of speech features on audio quality measures. Data from the mPower study suggests that phonation data from a bring-your-own-device setup is sufficiently consistent to predict motor disorders to some degree (see e.g. Lenain et al., 2020).
 1. Prosodic features as f0 range, speech rate, pause duration, etc., depends directly on the characteristics of the stimulus.

- We agree that this is the case. This is where completion of the same tasks in all clinical groups will be helpful, where cross-indication comparisons will be completed in the same task, or the same set of tasks.
 1. And finally, reading introduces a new variable: this is a skill very different from those related to the rest of the language tasks, and again, dependent on the level of schooling.
- We agree that reading, but also other cognitive tasks, will be influenced by education level. This is why education information is collected for all participants, and will enable education effects to be evaluated and controlled for, as we now specify in the 'statistical analysis plan' section.
- At the same time, we now specify in Table 2 that the reading task is employed primarily to evaluate pronunciation variation in different accents and dialects whilst keeping the language content the same.
 1. In short, although this point of the review may seem long, it can be summarised in a recommendation that can be incorporated into the work in a synthetic form: to expand the information in table 2, including a more precise description of the tasks (which should coincide with those in table 3), and adding two columns, one for expected outcomes and other for limitations of each task.
- We have adopted the reviewers suggestion of including more information in table 2. This now includes more precise descriptions of the tasks, and evidence from prior research relating to task performance in the key clinical indications.
- However, we have not included a separate section on limitations, since these are likely to be broad and likely to be, to some extent, overlapping between tasks. Instead, we have included more discussion of challenges of utilising remote cognitive assessment methods in the introduction. Whilst we take on board the limitations of certain tasks such as digit span with specific cut-off scores as described in Schroeder et al., 2012, the abbreviated digit span task administered in the current study is not intended to be used in isolation for identifying clinical groups, but rather one of a number of tests, which in combination may be sensitive to different clinical indications.

If statistics are used are they appropriate and described fully? Not entirely

The above remarks on outcomes (question 6) would have statistical implications. But it would also be useful to clarify some other details:

- Speech measures: what methods will be used to discriminate the most predictive markers? The expression "several methods from the fields of signal/speech processing and natural language processing" (pg. 10, lines 21-22) is somewhat imprecise.
- What kind of "audio representation" (pg. 10, line 26) will be used?

In the statistical analysis plan we now provide more detailed information relating to the acoustic and linguistic representations that will be extracted, along with relevant references for additional information.

8. Are the references up-to-date and appropriate? Not entirely

We have now included a large number of additional references throughout the manuscript, with the total count rising from 61 to [109]. We have included the following references suggested by the reviewer:

Patel, U. K., Anwar, A., Saleem, S., Malik, P., Rasul, B., Patel, K., ... & Arumathurai, K. (2021). Artificial intelligence as an emerging technology in the current care of neurological disorders. *Journal of neurology*, 268(5), 1623-1642.

Fraser, Meltzer, Rudzicz, Linguistic features identify Alzheimer's disease in narrative speech. J Alheimers dIS. 2016;49:407-22

Mueller, K. D., Hermann, B., Mecollari, J., & Turkstra, L. S. (2018). Connected speech and language in mild cognitive impairment and alzheimers disease. A__review of picture description tasks. Journal of clinical and experimental neuropsychology, 40(9), 917-939.

Cummins, N., Scherer, S., Krajewski, J., Schnieder, S., Epps, J., & Quatieri, T. F. (2015). A review of depression and suicide risk assessment using speech analysis. *Speech communication*, 71, 10-49.

9. Do the results address the research question or objective? N/A

10. Are they presented clearly? N/A

11. Are the discussion and conclusions justified by the results. N/A

12. Are the study limitations discussed adequately? No

The only paragraph discussing one of the limitations of the study, poor compliance, appears on p.10, l. 36-42, where some measures to address this potential issue are also mentioned.

Apart from this, the only acknowledged limitations are listed in pg. 2, along with the strenghts: "study design uniquely allows for investigation of the same tasks across a wide number of indications", "Limited sample size" and "Short follow-up period".

We have now provided more information on potential limitations throughout the manuscript, primarily in the 'methodological issues and limitations' section. Specific concerns are addressed below:

a) The type of stimuli used: effects of lexical frequency, grammatical complexity, argumentative structure, etc.

The ASRT parallel stimuli are balanced for grammatical complexity. See the supplementary materials in the Skirrow et al. (2022) preprint -

here: <https://www.medrxiv.org/content/10.1101/2021.10.12.21264879v3>. Whilst covering a variety of themes, the stories are structured similarly, involving a subject (a person) and their background, an event or challenge and its resolution. They are also constructed to contain a similar quantity of numbers and proper nouns). We cannot provide additional information on lexical frequency of the words included in these stories, since this has not been evaluated.

b) Effects of the channel used: telephone or videoconferencing platform.

Differences in the recording channel used (software, device hardware and microphone type) are likely to lead to differences in the audio data obtained. To evaluate how important these differences are, participants are recorded in two simultaneous ways during their virtual visits - via videoconferencing software and through a mobile phone application on their own smartphone devices. This will allow for direct comparisons to be made between the different platforms used. Whilst this cannot be considered an exhaustive comparison, it will allow us to quantify the effects of some of these differences and evaluate their likely importance to the results generated.

c) Effects of learning by repetition of tasks.

Practice and learning effects of parallel test variants over repeated assessment will be characterised with linear mixed effects models. This is now reflected in text.

d) Effects derived from the characteristics of the sample, which may be different in the neurodegenerative and psychiatric disorder groups.

We have provided more information in the ‘Statistical analysis plan’ section, in which we now describe how different confounding factors will be accounted for, for example, demographic variables such as age.

13. Is the supplementary reporting complete (e.g. trial registration; funding details; CONSORT, STROBE or PRISMA checklist)? Yes, but...

The instructions for reviewers of study protocols state that "Protocol papers should report planned or ongoing studies. The dates of the study should be included in the manuscript." This study clearly mentions that participants completed online questionnaires in a total of 12 days spread over four weeks, but I have not found the specific dates on which these tasks were carried out.

Study recruitment is ongoing - specific start and end dates for the study have been added to the text under ‘participants’. In terms of the follow-up dates for online questionnaires, there is flexibility as to when participants could complete these during the follow-up window and as such specific dates are not available for their completion.

14. To the best of your knowledge is the paper free from concerns over publication ethics (e.g. plagiarism, redundant publication, undeclared conflicts of interest)? Yes

15. Is the standard of written English acceptable for publication? Yes

References mentioned

We have greatly expanded the number and breadth of references in the manuscript. Of the specific references suggested to us, we have included the following:

Codina-Filbà, J., Escalera, S., Escudero, J., Antens, C., Buch-Cardona, P., & Farrús, M. (2021, June). Mobile eHealth platform for home monitoring of bipolar disorder. In *International Conference on Multimedia Modeling* (pp. 330-341). Springer, Cham.

Costa Beber, B. & Chaves, M. L. (2014). The basis and applications of the action fluency and action naming tasks. *Dementia & Neuropsychologia*, 8, 47-57.

De Marchi, F., Contaldi, E., Magistrelli, L., Cantello, R., Comi, C., & Mazzini, L. (2021). Telehealth in neurodegenerative diseases: opportunities and challenges for patients and physicians. *Brain Sciences*, 11(2), 237.

Omberg, L., Chaibub Neto, E., Perumal, T. M., Pratap, A., Tediario, A., Adams, J., ... & Mangravite, L. M. (2021). Remote smartphone monitoring of Parkinson’s disease and individual response to therapy. *Nature Biotechnology*, 1-8.

VERSION 2 – REVIEW

REVIEWER	Marrero Aguiar, Victoria Universidad Nacional de Educación a Distancia
REVIEW RETURNED	09-May-2022

GENERAL COMMENTS	The latest version of the paper overcomes, in my opinion, most of the weaknesses detected in the previous version. Table 2 now provides very relevant information, and the information added in the statistical analysis plan also helps to better explain the scope of the results. Undoubtedly, the increase in the literature consulted and cited has also led to a significant improvement of the article. I congratulate the authors for their effort and for the outcome.
---

REVIEWER	König, Alexandra
-----------------	------------------

	Institut National de Recherche en Informatique et en Automatique Centre de Recherche Sophia Antipolis Méditerranée
REVIEW RETURNED	17-May-2022

GENERAL COMMENTS	No additional comment to the author
-------------------------------------